# Review on Outbreak Dynamics, the Endemic Serotypes, and Diversified Topotypic Profiles of Foot and Mouth Disease Virus Isolates in Ethiopia from 2008 to 2018

**DOI:** 10.3390/v11111076

**Published:** 2019-11-18

**Authors:** Ashenafi Kiros Wubshet, Junfei Dai, Qian Li, Jie Zhang

**Affiliations:** 1State Key Laboratory of Veterinary Ethological Biology, National/OIE Foot and Mouth Disease Reference Laboratory, Lanzhou Veterinary Research Institute, Chinese Academy of Agricultural Sciences, Lanzhou 730046, China; 2Ethiopia Agricultural Research Council Secretariat, Addis Ababa 8115, Ethiopia

**Keywords:** FMD, Ethiopia, outbreak dynamics, serotypes, topotypes

## Abstract

Foot and mouth disease (FMD) endemicity in Ethiopia’s livestock remains an ongoing cause for economic concern, with new topotypes still arising even in previously unaffected areas. FMD outbreaks occur every year almost throughout the country. Understanding the outbreak dynamics, endemic serotypes, and lineage profiles of FMD in this country is very critical in designing control and prevention programs. For this, detailed information on outbreak dynamics in Ethiopia needs to be understood clearly. In this article, therefore, we review the spatial and temporal patterns and dynamics of FMD outbreaks from 2008 to 2018. The circulating serotypes and the topotypic profiles of the virus are also discussed. FMD outbreak data were obtained from; reports of MoARD (Ministry of Agriculture and Rural Development)/MoLF (Ministry of livestock and Fishery, NVI (National Veterinary Institute), and NAHDIC (National Animal Health Diagnostic and Investigation Center); published articles; MSc works; PhD theses; and documents from international organizations. To effectively control and prevent FMD outbreaks, animal health agencies should focus on building surveillance systems that can quickly identify and control ongoing outbreaks and implement efficient preventive measures.

## 1. Introduction

Foot and mouth disease virus played an important role in the histories of both human and veterinary medicine and to the field of virology in particular [1]. Some 500 years ago, a disease with a similar clinical manifestation to FMD disease was reported in Italy [2]. In the late 18th century, German virologists proved that foot and mouth disease virus was the first filterable animal virus [1]. Recently, FMD became a global issue mainly as a result of rapid genetic evolution, its contagious nature, various modes of transmission (direct contact, airborne, and via fomites) [3,4,5,6], and wide range of host preferences (more than 70 cloven-hooved animal species [7,8]. Controlling FMD is extremely resource intensive, which is mostly correlated with the convalescence in some cloven-hoofed animal species and long-term virus shedding from affected animals [1,8]. FMD afflicts livestock population in more than 80 countries [9]. These and other reasons possibly enable the disease to remain as a very important global concern.

The causative agent of FMD disease is *Aphthovirus*, a positive-sense single-stranded RNA virus with a very low molecular weight ranging from 7.2 to 8.4 kb [10]. It lacks an envelope and is grouped under the family of Picornaviridae, genus *Aphthovirus* [11]. It is 25 to 30 nm diameter in size, being a very simple and small in structure, which accelerates the air transmission of the virus, allowing it to spread over long distances in a very short time by following the nature of the wind speed and direction [12,13].

Through the epidemiological eyeglass, and from disease control perspectives, FMD weighs as seven immunological distinctive diseases, mainly due to the seven recognized serotypes currently circulating worldwide [14]. For this reason, immunity development by animals to one FMDV serotype does not protect them from other serotypes, and protection from other strains within a serotype varies with their antigenic similarity [15].

Animal species, breed, immunity status, and virus infection dose are some of the factors that affect the FMD infection rate [15]. Exposed animals could result in 100% morbidity [10,15]. In the majority of FMDV strains, the case fatality rate is higher in young animals (5% to 94% in lambs, 80% in some groups of calves, and 100% in suckling piglets) than adult livestock species (1–5%) [15,16,17]. 

The occurrence and economic influence of FMD differs throughout the world [18], because the disease varies markedly between FMD endemic and FMD non-endemic countries, developed and developing countries, and also among developing countries [19]. The outbreaks of this contagious disease can seriously affect the economy of a country in terms of production loss, export bans, vaccination costs, and losses from tourism in exposed regions [20,21,22]. For instance, annually, about 2.35 billion doses of FMD vaccines are administered to livestock throughout the world [11,23], and the total remittance is estimated to be about US$20.7 billion at its peak cost (US$9 per dose) [24]. 

In general, the economic impact of FMD is highest in Africa, China, and India [18]. In Africa in particular, despite its US$2.32 billion impact (from direct production losses and vaccination only), control of the disease is not yet prioritized, standard vaccination regimens are too costly, its economic impact is underestimated, and its epidemiology is not clearly understood [25].

Additionally, FMD is a disease of animal welfare concern due to the standard requirements for a massive culling of infected and potentially ’in contact’ animals when outbreaks occur in FMD-free regions [26]. Regardless of the rate of natural death from FMD, however, the economic impact when a country experiences an outbreak is made even more severe because of the need to quarantine and slaughter infected populations; in essence, a diagnosis of FMD may lead to culling of the entire affected populations [27]. 

The epidemicity of FMD in 2001 in the United Kingdom, which triggered a livestock culling campaign involving the slaughter of more than 6.5 million animals, is a very good example [28]. On the one hand, many countries like Japan, New Zealand, Australia, and Mexico remained free from FMD disease [29]. On the other hand, some countries considered free of FMD disease perhaps experience periodic FMD outbreaks and are obligated to maintain their capacity for rapid detection and control [30]. Some African countries have also been vigorously working to eradicate this devastating disease even though most of the states have no, or ineffective, control policies and programs. 

According to the recent research reports, six serotypes of FMD virus (O, A, Asia-1, SAT-1,-2, and -3) are circulating globally [31]. FMD outbreaks due to serotype C have not been reported in Africa since 1983 (Borena, Ethiopia) and 2004 (Kenya) nor in other parts of the world, such as in Europe, since 1989 (Italy), in South America since 2004 (Brazil), and in Asia since 1995 (India and the Philippines) or 1996 (Nepal) [32,33]. The genetic and antigenic divergence is a common feature among all FMDV serotypes. Serotype SAT2 comprises the broadest genetic topotypes [34,35].

Excluding Asia 1, all FMDV serotypes have been isolated in the African continent [36]. The main reasons for the epidemiological abundance and maintenance of the disease in the region are uncontrolled movement of domestic and wild animals and high quantities of persistently infected African buffaloes [37]. The presence of multiple FMDV serotypes circulating in the continent, therefore, results in periodic outbreaks.

Ethiopia is one of the FMD-endemic countries in the horn of Africa, with almost more than five serotypes prevailing so far. Epidemiological surveys in Ethiopia indicated that FMD outbreaks occur every year almost throughout the country. Some endemic part of Ethiopia, such as Borena, which is the main source for live animal exports, experiences multiple FMD outbreaks per year [38]. Such a high number of outbreaks has become a serious challenge for the livestock trade and meat industry of the country. Genetic characterization of FMD in Ethiopia from 1981 to 2007 provided comprehensive and summarized information about the outbreak dynamicity in the country [39]. However, recent dynamics and patterns of the disease outbreak and molecular epidemiology of FMDV is not well-documented. Hence, understanding the pattern of disease outbreak in various epidemiological situations in a different period can provide a framework for the prevention, control, and eradication of the disease. This review, therefore, aims to deliver details on FMD outbreak of the past 10 years (2008–2018) by profiling its temporal and spatial dynamics to fill the gap. It also summarizes the endemic serotypes and profiles of topotypes throughout this described periods.

## 2. Global View on FMD Serotypes and Strains 

The scientific community across the globe are convinced on the presence of genetic and antigenic divergence among FMDV serotypes regardless of how high and how less the serotypes are. 

### 2.1. FMDV Type O 

This type is the most extensively studied and common FMD serotype in the globe [40]. It comprises eight topotypes [41], including: 1. Cathay; 2. Middle East-South Asia (ME-SA); 3. South-East Asia (SEA); 4. Europe-South; 5. America (Euro-/SA); 6. Indonesia-1 (ISA-1); 7. Indonesia-2 (ISA-2); and 8. East Africa (EA) and West Africa (WA). Except topotype ISA-1 and ISA-2, which are solely restricted to Indonesia, regardless of their naming, almost all of these topotypes have been reported in different parts of the world [41].

Tekleghiorghis et al. [37] indicated that East Africa (EA1–4), Middle East-South Asia (ME-SA), and West Africa (WA) types were the most common circulating topotypes in Africa from 1990 to 2011. Serotype EA-3 and EA-4 were particularly identified in Ethiopia at different periods [42]. This serotype contains five neutralizing antigenic sites on the external surface; the G-H loop forms the most prominent surface of viral prorotein1 (VP1) and the carboxylic end of VP1 contributes to the antigenic site 1 with critical residues at positions 144, 148, 154, and 208 [42].

### 2.2. FMDV Type A 

Serological type A FMD virus is often considered to be the most antigenically diverse Eurasian serotype and has been emerging with antigenically novel strains, particularly in western Asia, which lacks cross-protection between them [40]. Serotype A is broadly categorized into three distinct topotypes (AFRICA, ASIA, and EUROPE-SOUTH AMERICA (EURO-SA) [43]. Recombination in serotype A occurs much more than in the other serotypes [44,45]. Briefly, in serotype A, the four antigenic sites reported are found to be in similar positions to that of serotype O unlike antigenic site 3. Two prominent antigenic sites were described on VP1 (residues 140–160) coupled with two minor antigenic sites on VP1 (residue 169) and the C terminus of VP1. The second antigenic site is located in VP2 at residue positions 72 and 79 [46].

### 2.3. FMDV Type SATs 

South African types have significantly higher sequence diversity within each other than in serotype O [34]. SAT1 consists of eight geographically highly localized topotypes, and SAT-2 showed higher genetic diversity, with a total of 14 topotypes and three serological subtypes, with five of these possibly extinct [34,35,47].

Intratypic variation is more common in SAT-types than European serotypes (O, A, and C) [48]. The variation can be at the nucleotide and amino acid levels. At the nucleotide level, 34%, 40.4%, and 36.1%, and at amino acid levels 25.4%, 27.5%, and 24.1% for SAT-1, -2, and -3, respectively [48].

Additionally, SAT-3 has relatively less epidemiological coverage on the continent and is rarely isolated from African buffalo. It contains six topotypes with 25 genotypes, where four of them found in southern Africa, and two were unique to East Africa [34]. The two antigenic sites of SAT2 are located in the G-H loop of VP1, in the lower stream coding region to a diversified sero-active protein motif (RGD), at amino acid position 147, 148, 156, and 158 and 154 [49].

## 3. FMDV Serotypes Status on Sub-Sahara-African 

Africa is the foundation for FMD SAT serotypes and as a continent has also been taking the lion share of the maintenance of all FMD serotypes except Asian origin. Ethiopia immensely contributes to this share. Within Africa, there is a marked regional difference in the distribution and prevalence of these serotypes and their intra-typic variants [32,50]. Whereas Asia contends with four serotypes (O, A, C, Asia-1), and South America is restricted to only three (O, A, C) [32].

FMDV serotypes O, A, and the South African Territories (SATs) are the most circulated serotypes in the continent. Serotype O is the most widely distributed in eastern and western Africa followed by A, while SATs virus is mostly found in the southern region [26]. Even though the distribution of SAT viruses is restricted to Sub-Saharan Africa, some incursions of SAT1 have been reported in Greece while both SAT1 and SAT2 have occurred in the Middle East [32,51,52]. 

The Office International des Epizooties (OIE) outbreak profile in Africa from 2000 to 2010 showed that SAT2 was drawn as a predominant serotype (41%) followed by the northern France origin O serotype (23%) [53]. Multi topotype SAT 2 endemicity and outbreaks out of the Sub-Saharan terrestrial ranges have also been observed in countries south of the Sahara desert, and the Northern African and the Middle East region [54], such as Libya, Egypt, Palestinian Autonomous Territories (PAT), and Bahrain [54].

The earliest incursion of SAT2 in Northern Africa and the Middle East has become a potential threat to European nations in the Mediterranean Basin and mainly FMD-free countries densely populated by pigs [53]. Inoculated pigs could become infected with a SAT2 serotype according to Mouton et al. [54]. 

Isolation of the FMDV serotype C in Kenya during 2004 in cattle was suggested as a re-introduction of the vaccine strain into the field [33]. On top of this, phylogenetic analysis study by Ayelet et al. [39] showed that all serotype C virus isolates in Africa are found in single lineage. The FMDV burden circulating around the globe has been manageably grouped into seven ‘localized pools’ (Figure 1) in consideration of the genetic and antigenic characteristics of virus strains and the estimation of the FMD outbreak status in a specific region [52,55,56,57]. Generally, Africa has been divided into three FMDV pools: East Africa (pool 4) with serotypes O, A, SAT-1, SAT-2, and SAT-3; West Africa (pool 5) with serotypes O, A, SAT-1, and SAT-2; and southern Africa (pool 6) with serotypes SAT-1, SAT-2, and SAT3 [58]. A different report has indicated 6 topotypes identified for serotype O; 2 for serotype A; 3 for C; and 9, 14, and 5 topotypes for SAT-1, SAT-2, and SAT-3, respectively [59]. 

Co-occurrence of virus serotypes and topotypes is a common phenomenon between East and West African geographical settings, which belong to pool 4 and pool 5, respectively. Serotype O (topotype EA-3) incidence in Niger (2007) and Nigeria (pool 5) were genetically related to the serotype and topotype virus isolated in East Africa, Eritrea (2004 and 2011), in Ethiopia (2005, 2006, 2008, and 2010–2012) and in Sudan (2005 and 2008–2011) (pool 4) [26]. SAT-2 topotype VII is similarly shown in these two pools [37]. The majority of the disease-free areas in the continent are found in the southern region (like parts of Botswana, Namibia, and South Africa), where cordon fencing and regular vaccination is affordably implemented as FMD control strategies [37,60]. 

Southern African countries practice more industrialized farming systems, making control of FMD by mass and regular vaccination affordable [25]. Hence, in this region, FMD is endemic only in African buffalo (*Syncerus caffer*), which is a potential source of foot and mouth disease virus (FMDV) for livestock [25,48,61]. In the Eastern region of the continent, on the other hand, the farming system is more traditional, the role of wildlife in the epidemiology of FMD is not much clear, and massive animal movement is practiced [25]. Because of these and other factors, many countries in Africa have not been able to create FMD-free zones and hence, could not meet the OIE and WTO requirements for the international trade of live animals and animal products [26]. Perhaps, it is practically complex and difficult to eradicate FMD in sub-Saharan Africa unlike in North America and Western Europe, mainly because of the ever-present threat of the reservoir host, the African buffalo, lack of animal movement control supported by law, and limited practices of vaccination programs [62,63], in addition to the extensive nature of the farming system.

## 4. Endemic Serotypes of FMDV in Ethiopia 

The history of foot and mouth disease in Ethiopia accounts more than six decades from now [39]. Serotype C and O were the first identified FMD serotype in Ethiopia [64,65]. The establishment of three African origin (SAT1-3) FMDV serotypes in the country, unlike O and C, has taken many years (22 years), regardless of the epidemiological proximity.

During the early periods of FMDV entry into the country, only a small percentage of outbreaks were reported and typed. As a result, serotypes A and SAT 2 were identified in 1969 and 1989, respectively [64,66]. During 1988 to 1991, serotyping of FMDV started massively at the national veterinary institute (NVI) and National Animal Health Diagnostic and Investigation Center (NAHDIC) [41]. National FMD outbreaks investigations from 1974 to 2007 resulted in isolation of serotypes O, A, C, SAT1, and SAT2 [50,67,68,69]. The antibodies of SAT2 were detected in sera collected from cattle populations of north Omo, the south-western part of Ethiopia [66], but its territorial counterpart SAT1 was isolated from buffalo and cattle in 2007 for the first time. The temporal and spatial distribution of FMD in Ethiopia is very complex, mainly due to the presence of diverse serotypes (O, A, SAT 1, and SAT 2) and the presence of widely susceptible host species (cattle, sheep, goats, and pig) [70] in addition to the presence of wild reservoirs like buffalo [38].

A serological survey in wildlife indicated that antibodies against A, SAT1, and SAT2 FMD types were detected from buffalo serum collected at Omo national park [38]. African buffalo, which are largely found in the Mago and Omo national parks, are potential FMDV reservoirs. Sahle et al. [38] also explained that uncontrolled cross-border animal movement to neighboring countries, such as from the Borana pastoral area to Kenya and vice versa, make Ethiopia the only country in its continent where five serotypes of FMD virus (O, A, C, SAT1, and SAT2) were circulating. Four (O, A, SAT 2, and SAT 1) out of the seven serotypes of FMDV remain mostly prevalent serotypes in Ethiopia for the last 20 years [32,50]. Ayelet et al. [39] added that O and A are the most dominant serotype, covering about 72% and 19.5%, respectively. The spatial distribution of the serotypes in Ethiopia (Map) is illustrated in Figure 2.

## 5. FMD Outbreak Dynamics in Ethiopia from 2008 to 2018 

In this review, we considered the outbreaks recorded by NAHDIC, NVI, (MoLF /MoA), reports from articles published in reputable journals, MSc works, PhD theses, and reports from international organizations in the period from 2008 to 2018. We perceived variations in FMD outbreak reports in different studies. This might be related to the coverage of the study area, limited sample analysis, shortage of adequate information, and priorities of investigators. Hence, we attempted to address the outbreaks that occurred in the periods from 2008 to 2018. 

It was noticed that the incidence of FMD outbreaks in Ethiopia is growing and is visibly raised in all regions of Ethiopia, starting from 1991 [38]. In 1999, approximately 10% of cattle were under risk of FMD viral infection, and consequently outbreaks increased rapidly in 2000 and 2001 from 27 to 88 [67]. However, they sharply decreased from a total of 176 outbreaks in 2001 to 26 in 2005 [72]. 

Ministry of Agriculture and Rural Development outbreak studies report that from 2002 to 2006 FMD outbreaks occurred each year in Ethiopia, with the highest number of outbreaks registered in 2004 (134 outbreaks), and observed more often in the North Shoa zones of both the Oromia and Amhara regions [73,74]. 

FMD outbreaks affected roughly 30% of districts in the country every year, with different degrees of economic impacts [31,39,73]. The number of FMD outbreak records in 2008 to 2018 [75] was 4.33 times higher than outbreak reports in 1981 to 2007 [39]. 

A total FMD of 72 outbreaks was reported in the periods from 2008 to 2009 mostly in Oromia (48 outbreaks) followed by Amhara (10 outbreaks), Addis Ababa (5 outbreaks), and Tigray (4 outbreaks) [76]. However, at the district level, the seroprevalence was highest (41.5%) in the Eastern zone of Tigray region (lower in overall prevalence) followed by the Guji zone of Oromia and Yeka district of the city of Addis Ababa, with 32.7% and 30%, respectively, as shown in Table 1 [76]. In the same year, Negusssie et al. [69] reported a total of eight FMD outbreaks in Amhara, Oromia, and Addis Ababa.

The serological study results of Ayelet [73] showed that only Gambella and Benshangule Gumuz were found to be serologically FMD free and were considered to be hypothetical candidates for the establishment of FMD-free zones in the country. Other seroprevalence studies on FMDV from 2007 to 2011 in diverse ecological regions of the country reported 9% to 26% and 48%, both at an animal and herd level, respectively, [65,69,77,82,83,84].

Ministry of livestock and fisheries case book records from 2009 to 2018 show that a total of 884 outbreaks occurred annually at the national level [75]. In line with this, the highest proportion (438/884) of outbreaks was reported from the Oromia regional state and nearly one-third from the total of the outbreaks were in 2012, but the lowest was in 2014 [4]. 

The ministerial office believed that these prevalence may not accurately explain the epidemiological situation of the disease in the country due to the fact that the insidious spread of the disease and sub-reports from livestock owners [75]. In the year 2014 to 2017, a total of 35 outbreaks were recorded throughout the country, with the highest outbreak number again in the Oromia region [75]. The same report compiled an overall seroprevalence of 20.79%, with the highest prevalence in the Benshangul region (which previously experienced a much lesser incidence) followed by SNNP, Oromia, and Gambella, respectively. However, the FMD seroprevalence recorded in Tigray (highest in the previous period) was the lowest in this study report. Therefore, on the basis of the ministry of livestock and fishery reports, various seasons of the years demonstrate quite different degrees of outbreaks, as illustrated in Figure 3 and Figure 4 [75]. Also, seasonal analysis roughly implicated that the highest FMD outbreaks occurred from December to early February, but this was lowest in April. The reason why the peak occurred during these dry seasons of the years might be associated with factors, such as drought. This is because in these dry seasons, particularly, pastoralists used to move their herds over long distances to find pasture and water, which accelerates transmission of the virus [68,77,80,83]. 

FMD outbreak frequencies exhibited variation across the different farming systems in the country [31,85], and was significantly higher in the marginal lowland areas of the country than highlanders with integrated crop livestock farming systems (CLM) [39]. Besides, in Jemberu et al.’s study [31], the herd- and individual-level morbidity rate of FMD was 10 times higher in pastoralist than CLM. Similarly, studies in different periods in the pastoral area indicated that Borena cattle experienced an FMD prevalence of 14.8% [73], 26% [77], and 21% [38]. Similarly, studies in different periods in the pastoral area indicated that Borena cattle experienced an FMD prevalence of 14.8% [73], 26% [22], and 21% [76]. This is mainly associated with the majority of illegal livestock trade and uncontrolled border movement of animals. Hence, livestock movement (both formal and informal) to these borders from every corner plays an essential role in the spread and epidemiology of FMD [36,38]. However, in recent times, the incidence of this disease has also been increased in the highland areas, where more than 60% of the total cattle population exists [31].

Retrospective data (2006–2016) on FMD outbreaks at the national level by NAHDIC showed 12.6% and 58.2% at the individual and herd level, respectively [86]. In the last 10-year (2008–2018) report, serotype O was mostly identified followed by serotypes SAT 2 and A, whereas SAT 1 serotype occurred for the first time in the areas bordering Ethiopia near Sudan. Similarly, during the last 20-year (1981–2007) outbreak reports, serotype O was dominant, followed by A, but SAT2 broke out after a 16-year gaps [39]. The prevalence of FMD in different agroecological situations across the different regional states of Ethiopia is summarized in Table 1. 

### Topotypic Profiles of FMDV Serotypes in Ethiopia from 2008 to 2018

Regular monitoring of the circulating topotype and lineages of FMDV serotypes in livestock may possibly be important for the selection of appropriate vaccine strains for strategic control and prevention practices [38]. Molecular epidemiology investigation of FMD were mostly carried out by phylogenetic analysis on isolates from outbreaks based on their virus protein (VP 1) region [38,50,87]. Based on phylogenetic analysis studies, Ethiopia shared several topotypes with different countries across the globe, and is the only origin for some topotypes [39]. 

In an outbreak study from 1981 through to 2007, serotype O and SAT1 reemerged with a new topotype EA4 and IX, respectively. From the year 2008 to 2018, topotypes included EA-3 and EA-4 within serotype O; lineage G-VII (Africa topotype) within serotype A; IV, XIII, and XIV within serotype SAT 2; and IX within serotype SAT 1 [31]; and all topotypes are summarized in Table 2. 

## 6. Estimates of EA3 and SAT2 Viral Movements 

Apart from globally well-known regional FMD epidemic pools, recently, epidemic jumps have been shown between different pools where serotypes from one pool have started to emerge into specific serotype-free regions [24]. The most recent example is the FMDV SAT 2 endemicity out of the Sub-Saharan terrestrial range, in countries nearly to the south of the Sahara desert, and the Northern African and Middle East region [88]. Egypt claimed the transmission FMD from Ethiopia to its herd in 2007, and ceased importing live cattle from 2007 to 2010 [71]. Likewise, an FMDV SAT serotype sporadic outbreak in Saudi Arabia and Kuwait in 2000 is estimated possibly due to the movement of the animals from Africa to the Middle East and Asia [89].

Unlike many other countries, Ethiopia is in the middle of three pools (pool 5, 4, 3), which are connected with southern Africa’s territories via pool 4 and to the Middle East via pool 3 [90,91]. Hence, the WRFMDL (2010–2017) report in (Figure 5) indicated that FMD outbreaks of the EA3 viral lineage of serotype O and SAT2 VII reports from East Africa have been transported to the Middle East countries found in pool 3. This long transpooling of the serotypes virus is probably associated with animal trade and the genetic variability of the virus strains. 

## 7. National Outbreak Investigation and Reporting System 

Since very recent times, FMD outbreaks in Ethiopia have been most commonly investigated in two laboratories: The National Animal Health diagnostic and Investigation Center (NAHDIC), which is an East African reference laboratory for many diseases, and the National Veterinary Institute (NVI), veterinary vaccine producing center for the nation. These two institutions in coordination with the sub-national veterinary laboratories are the backbone of outbreak investigation and the implementation of disease prevention and control activities in the country. Within the regional laboratory network, NAHDIC technical teams are first to be directly involved in the outbreak area together with a local technical team participating in investigation and control measures. NAHDIC receives outbreak samples and performs virus isolation and virus neutralization tests, and RT-PCR. In addition, Sandwich ELISA and other serological tests (liquid phase blocking ELISA and 3ABC ELISA Test) are also conducted to detect antigen or antibody of FMDV in the serum sample. World Reference Laboratory (WRL) at Pirbright institute has been working with NAHDIC for the confirmation of samples, for further genetic characterization and vaccine matching. Although OIE member countries have an agreement to report each outbreak through OIE pathways, the organization (in some of its reports) claimed inadequate and untimely recording and notification of FMD disease outbreak. This may be related to the fears of exporter countries regarding their trade banning issues. On the other hand, it is also suggested that this may be due to a comparatively high tolerance of local breeds to the clinical episodes of the disease [92]. Ethiopia’s successive FMD outbreak reports consisting of five serotypes was recorded in the webpage of WRFMDL as indicated below in Table 3 [43].

## 8. National FMD Control Attempts, Approaches, and Perspectives 

Applicable control measures for FMD involve animal movement restrictions, a vaccination program, animal quarantine, environmental sanitary controls, outbreak investigation, serological surveillance, and slaughtering of sick animals [94]. In Ethiopia, FMD outbreaks have remained an economical burden to farmers due to the inaccessibility of the vaccine in particular, and lack of awareness about the vaccination program [77]. Prophylactive vaccination is very limitedly practiced and ring vaccination is carried out in outbreak areas to limit further spread of the virus [39]. Trivalent FMD vaccines against FMD strain O, A, and SAT2, particularly topotypes of EA 3, Africa 3, and XIII [90], respectively, are commonly produced and distributed in the country, regardless of the quantity and quality. Most often, the vaccines are insufficient relative to the livestock population in the country, and affordability questions in mind, these influential farmers and pastoralists lose interest in using the vaccines. Until recently, only centrally coordinated vaccination activities have been carried out in some market-oriented dairy farms and feedlots in urban and peri-urban areas [95]. In fact, these vaccination efforts have either been reactive vaccinations in response to outbreaks or regular preventive vaccination [96]. In addition, livestock producers in the country use palliative antibiotics or traditional treatments to monitor the clinical signs of FMD in cattle [31]. Despite several efforts and attempts to design an FMD control strategy at the national level, an officially endorsed control plan for FMD has not been established. Recently, Ethiopia has joined the progressive control pathway (PCP-FMD) network, launched by FAO and OIE in Bangkok in June 2012 [97], and has started implementing this since 2017 with the progress reaching stage one [71].

Thus, commitments and efforts are undergoing as a long-term strategic approach of FMD control plans by adopting a progressive control pathway and short-term plan by the implementation of critical prevention and control mechanisms across baseline mapping, surveillance, and biosecurity to fulfill Ethiopia’s potential for livestock and specifically cattle export and to join the lucrative market. 

## 9. Conclusions 

Foot and mouth disease is endemic in Ethiopia and is a major obstacle to the development of the livestock industry as it imparts adverse effects on livestock production and exports. Outbreaks of FMD have shown periodical dynamics under different circumstances. FMD outbreaks appear every year in almost any corner of the country under diversified geographical settings and with measurable differences in the magnitude of the incidences. The epidemiology of FMD in Ethiopia is more problematic mainly due to the survivor of the virus in a wide host range (domestic and wild host species) as well as roaming freely across borders between neighboring countries, communal animal grazing and watering points, and lacking the control of animal movements. Furthermore, the lack of economically available and environmentally stable prophylactic vaccination and veterinary infrastructure to handle the dynamic pattern of the outbreaks on a large scale each year dramatically contributes to the frequent occurrence of the disease. Consequently, this makes FMD prevention and control extremely challenging. Hence, an organized national FMD control strategic plan is highly required. Besides, diagnostic technology and control strategy for Foot and Mouth Disease in Ethiopia should be reviewed, and every gap needs to be identified to take considerable and effective measures in strengthening the present efforts on implementing the strategic fight against such a momentous disease in the country. Accurate epidemiological models are required as well as the formulation and development of effective control policies, ultimately for planning and outbreak preparedness in the country in broader arrays. 

## Figures and Tables

**Figure 1 viruses-11-01076-f001:**
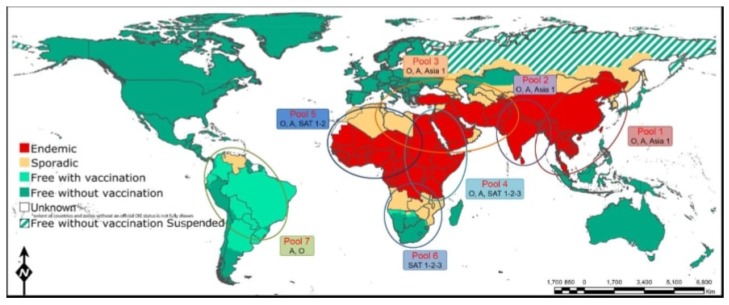
Mapping of the seven global endemic pools with estimated FMD outbreak status during 2017. Ethiopia is found in the middle of three pools (pool 5, 4, 3) and is connected to the Southern Africa’s territories via pool 4 and to the Middle East via pool (3). There is overlapping between pools 4 and 5. Virus circulation and evolution within these pools helps in estimation the jumping of virus between pools and for appropriately adapted vaccines. Map adopted from OIE/FAO Foot-and-Mouth Disease Reference Laboratory Network Annual Report 2017 [55].

**Figure 2 viruses-11-01076-f002:**
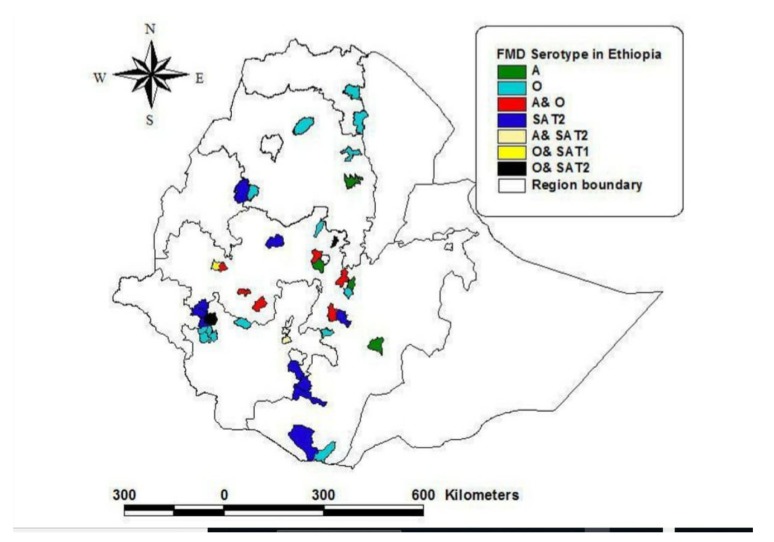
The figure shows the spatial distribution of FMD serotypes identified from outbreaks (2014–2017) in regional states of Ethiopia; O, A, SAT1 and SAT2 in Oromia and Addis Ababa, O, A and SAT2 in Amhara and Tigray, and only O serotype was identified in SNNPR, Somali, Benshangual and Afar. (The map was adopted from Daniel Gizaw presentation EU-FMD [71].

**Figure 3 viruses-11-01076-f003:**
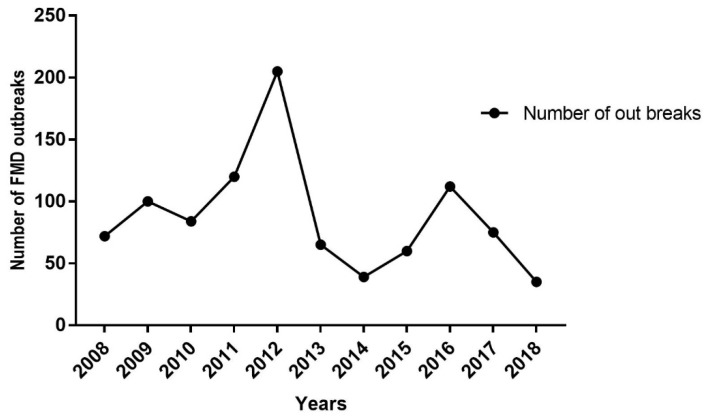
Numbers of FMD outbreaks (2008–2018) reports from different part of the country. The graph depicted that the highest outbreak recorded in 2012.The data out the outbreak was mostly retrived from MoLF (Ministry of Livestock and Fishery) and different research works [75,76].

**Figure 4 viruses-11-01076-f004:**
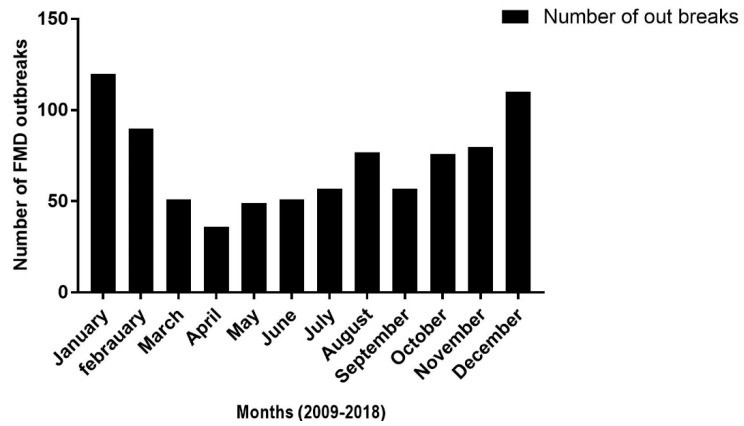
The average number of FMD outbreaks (2009–2018) by month in different parts of the country. This illustration shows seasonal dynamics with higher FMD outbreaks burdens occurred during peak dry season (from December to early February). The data of the outbreak was maintained from MoLF (Ministry of Livestock and Fishery) [75].

**Figure 5 viruses-11-01076-f005:**
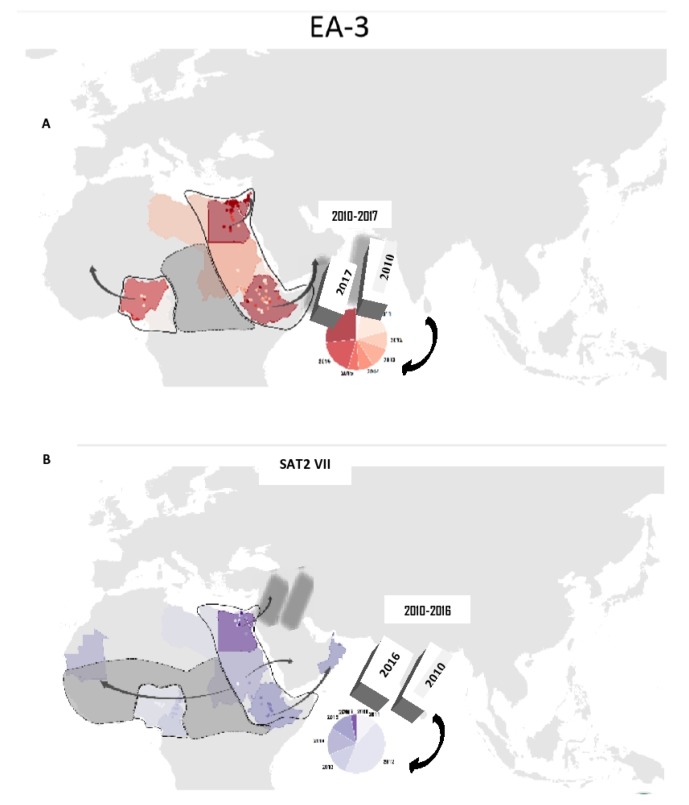
Conjectured transpooling of Serotypes of FMDV lineages between countries in different pools from year 2010 -2017. The pie diagram in (A) Shows the transpooling of serotype O/EA-3 lineage from east Africa zone at different periods including the new FMD outbreaks from O/EA-3 in palestein and isreal 2017. (B)Shows the spread of serotype SAT2 (topotype VII) FMDV lineage from east Africa in to North Africa and Middle East. Map adopted from OIE/FAO Foot-and-Mouth Disease Reference Laboratory Network Annual Report 2017 [55].

**Table 1 viruses-11-01076-t001:** Seroprevalence of FMD during the period 2006–2014 in eight regional states and an agrocological different regional state.

Regional States	Outbreak Investigation Area	Tested Animal Populations	Prevalence %	Cited
Oromia	Borena	134	55.6	[77]
	Moyale	174	16.1	[77]
	Adma-Mojo livestock export	4321	12.5	[20]
Addis Ababa	Yeka	40	30	[73]
	Bole	40	12.5	[78]
Tigray	Central zone	139	26.6	[73]
	Eastern zone	41	41.5	[73]
	Western zone	195	16.9	[79]
	Southern zone	75	24	[79]
Amhara	South achefer	101	52.5	[69]
	Habru	218	38.7	[69]
	Dangela	104	43.3	[69]
SNPPR	Hammer	104	13.5	[80]
	Arbaminch	90	7.3	[81]
	Jinka	162	4.9	[80]
	Semen bench	153	5.8	[39]
Afar	Zone 4	299	4.5	[73]
Somalia	Awabere	225	14.2	[82]
	Babile	159	15.1	[82]

**Table 2 viruses-11-01076-t002:** Serotype-based FMD outbreak reports from Ethiopia to WRFMDL in the year 1957 to 2018—the data were extracted from the World Reference Laboratory for Foot-and-Mouth Disease (WRLFMD).

Serotypes	FMD Outbreak Reported Years
Untyped	1998, 2000–2001, 2004–2006, 2008–2012
O	1957, 1961–1963, 1966, 1969, 1989, 1996, 2003–2018
A	1969, 1981, 2000–2002, 2008–2009, 2015, 2017–2018
C	1957, 1971, 1983
SAT 1	2007
SAT 2	1989–1991, 2007, 2009–2010, 2014–2015, 2018

Source [43].

**Table 3 viruses-11-01076-t003:** Distinct topotypes of the four dominant FMD serotypes in Ethiopia during the periods from 2008 to 2018.

Serotype	Topotype	Genotypes	Reference
O	EA-3		[39,56,91]
	EA-4		[39,91,93]
A	Africa	G-VII	[69,91]
		G-IV	[50]
SAT1	IX		[39,91]
SAT2	VII		[50]
	XIII		[39,91]
	XIV		[39]

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
