# Peer review of "Review on Outbreak Dynamics, the Endemic Serotypes, and Diversified Topotypic Profiles of Foot and Mouth Disease Virus Isolates in Ethiopia from 2008 to 2018"

_viruses, 2019, doi:10.3390/v11111076_

Round 1

Reviewer 1 Report

October 28th, 2019

Review: viruses-637592

“Review on- Outbreak dynamics, the endemic serotypes and diversified topotypic profiles of Foot-and-Mouth Disease Virus isolates in Ethiopia from 2008-2018”.

In this new version of the review article the authors to summarize the FMD outbreak situation in Ethiopia during the period of 2008-2018. The topic is of great interest since understanding the situation of any particular region affected by FMD leads the pathway to control and eventually eradicate the disease. The new manuscript has a very informative easy flow that allows the reader to understand the situation in the regions and possible consequences derived from the current circumstances. There are still several sentences that need further editing in order to make sense. Also, the quality if the figures need to be improved before the manuscript is ready for publication.

Specific comments:

1.     Sentence on line 39-40 (very smaller in size that range from 25-30 nm) needs to be re-written.

2.     Line 44: ‘circulating in worldwide’ should read ‘circulating worldwide’

3.     Line 79: ‘in other part of the world’ should read ‘in other parts of the world’

4.     Line 82: ‘Serotype SAT2 is frontier in terms of genetic topotypes” does not make sense. Please rewrite

5.     Line 90: ‘In some endemic part of Ethiopia’ should read’ Some endemic parts of Ethiopia’

6.     Line 93: remove ‘a’

7.     Lin 96: ‘FMDV lacks well-documented article” should read ‘FMDV is not well-documented’

8.     Line 97: ‘situations at different period’ should read ‘situations at a different period’

9.     Line 100: ‘throughout these described periods’ should read ‘throughout this described period’

10.  Line 110: ‘reported from different’ should read ‘reported on different’

11.  Line 119: ‘virus often considered to be antigenically the most diverse’ should read ‘virus is often considered to be the most antigenically diverse’

12.  Line 131: remove ‘in’

13.  Line 134: ‘relatively has less epidemiological’ should read ‘has relatively less epidemiological’

14.  Line 154: ‘becoming’ should read ‘become’

15.  Sentence in lines 285-288 should be re-written

16.  Sentence in line 292-293 should be re-written

17.  Line 298: ‘Ethiopia in the middle’ should read ‘Ethiopia IS in the middle’

18.  Section 7 could be included in section 5 where it seems more relevant and would be less repetitive.

19.  Line 331: ‘also conduct to’ should read ‘are also conducted to’

20.  Line 336: ‘This is may be related’ should read ‘This may be related’

Author Response

We have attached the comments and responses here below. We have also prepare the tracking for comments in the document with red and blue color. We are also prepare the revised final document separately. Figure 3 had some typographical error and we have corrected it already.

Reviewer 2 Report

The revised version of the manuscript entitled “Review on-Outbreak dynamics, the endemic serotypes and diversified topotypic profiles of Foot-and-Mouth Disease Virus isolates in Ethiopia from 2007-2018” has been improved, however, there are still some minor revisions to be addressed:

Line 206: During a serological survey, you can’t say that the virus has been isolated, but there is only evidence of the presence of type-specific antibodies.

Line 235-241: the authors start the paragraph listing the outbreaks that occurred in Ethiopia during 2008-2009 asserting that they are shown in Table 1, where on the contrary are reported the results of the seroprevalence during the years 2006-2014.

As far as I understand the Tigray district has had the highest seroprevalence during 2008-2009 but the lower number of outbreaks reported, in the parentheses, the authors use the term “prevalence” and it is confusing.

Table 1 caption: as far as I understand the data are referring to the seroprevalence, if so please include the term “seroprevalence” in the table caption instead of the term “prevalence”.

Line 260: As expressed by the authors in the same paragraph Tigray has had the highest seroprevalence and the lower outbreaks record during 2008-2009. I do not understand if in this case, the authors are speaking about seroprevalence or number of outbreaks.

Line 330: Sandwich ELISA is not usually a serological test, and serological tests are conducted to detect antibodies and not the antigen. Furthermore “ABC ELISA” is “3ABC ELISA”. Please rephrase the concept from line 330 to 331 because it is a little confusing.

Paragraph 9: The authors assert that a trivalent vaccine against FMDV serotypes O, A, and SAT2 are produced and distributed in the country, did the authors consider the seroprevalence data in respect to the presence of vaccination-induced antibodies?

Minor remarks:

Paragraph 2.2 FMDV type A: to be consistent with the description of the other FMDV serotypes would be better to include the number of topotypes in the description of Type A.

Line 167-168: the number of topotypes for each serotype has been described in the previous paragraphs.

Line 239: what does it means “similar year”, maybe “same years” is better.

Lines 240 and 345: outbreaks is written wrongly.

Line 247: “Another seroprevalence studies” is better “Other seroprevalence studies”.

Line 309: Include the reference: “Knowles N., Wadsworth J., Bachanek-Bankowska K., King D. (2016). VP1 sequencing protocol for foot-and-mouth disease virus molecular epidemiology”, Revue Scientifique et Technique de l'OIE. 35. 741-755. 10.20506/rst.35.3.2565.

Line 314: G VII is a lineage.

Line 326: the word "controls" is written wrongly.

Line 339: Table 2 is above not below.

Line 349: the topotype A in the vaccine composition is not correct.

Author Response

We would like to inform you that the responses for your comments are summarized in the PDF document attached below. We have prepared the comment tracking document (blue and red) and final revised manuscript ( with all these edition). We will attach the two documents in to your email. Because here we can not find place to upload. We also made some typographical  correction on the Fig3. And we wish to upload this Fig too.  

This manuscript is a resubmission of an earlier submission. The following is a list of the peer review reports and author responses from that submission.

Round 1

Reviewer 1 Report

The manuscript “Review on-Outbreak dynamics, the endemic serotypes and diversified topotypic profiles of Foot-and-Mouth Disease Virus isolates in Ethiopia from 2007-2018” has as objective the description of the FMDV circulation in Ethiopia from 2007 till  2018.

Besides the topic is interesting the manuscript needs a lot of adjustments:

ü  The manuscript does not allow to a smooth reading. The use of the English language should be revised. In many occasions it is difficult to understand what the author wants to say.

ü  The first paragraphs about general information on FMDV are not well organized and very long considering the main topic of the review, namely the FMDV situation in Ethiopia. In fact only a few paragraphs are dedicated to Ethiopia in specific.

ü  FMDV movements and dynamics within Ethiopia are not well supported by figures, figure 5 is not clear; in addition figure 5 and 4 are both from WRLFMD reports with no reference indication.

Unfortunately, for the reasons mentioned above, I think that the review should be rejected by the editor.

Reviewer 2 Report

Title: Review

 Review on- Outbreak dynamics, the endemic serotypes and diversified topotypic profiles of Foot and-Mouth Disease Virus isolates in Ethiopia from 2007-2018.

The authors provide a comprehensiveinformation on FMD outbreak dynamics in Ethiopia over ten years from (2007- 2018) in different regions andat different seasons, and summarized the endemic serotypes and profiles of topotypes within the time indicated.

The manuscript give a good overview about the dynamic of FMDV in east Africa and provide useful information about the link between the incursion of different serotypes in the middle east through Ethiopia,  however, I think the manuscript presentation can be improved significantly.  

The main concern in this manuscript is as follows:

The Abstract is not easy to follow, it can be re-written in a simple English.  

Line 42- re write this sentence in a simple English  

Line 46- replace light with small

Lines 48-50- long sentence, divide it into two sentence

Line 77 is missing and

Lines 78 to 80 long sentence.

Line 85 is missing (or) more

Line 90 to91- be consistent with the references styles currently you have two different styles. And update the reference numbering after all.

Line 97 there is one number missing in year 201?

Line 108, Antigenic should be lower case antigenic

Line 122 , delete and  (147, 148, 156, and 158 and 154)

Be consistent with the reference brackets you have ( ),  [ ] and sometimes names

Line 143- reference is missing the year of publication.

Line 146 you mentioned in the introduction line 74 that type C was last reported in 1995 please update.

Line 189 in is missing in Ethiopia

Line 191- reference is missing the year of publication

Line 196 – be consistent with the brackets - covers 72% and (19.5%)

Lines 211-215- Can you expand a bit about to include which serological tests were used e.g NSP or SP ELISA? Was the vaccination program active in this region?

Fig 1, data from 2007 is missing as you indicated in the abstract that this review is covering from 2007 to 2018.

Fig 3 is not clear. Explain did you produce this data. Is this dynamic represent any particular year?

Line 263 to 273 – This paragraph is not needed

Fig 5 try to improve the quality of the figure, you need a reference or method to describe how this figure was generated.

Reviewer 3 Report

July 26th, 2019

Review: viruses-557849

“Review on- Outbreak dynamics, the endemic 3 serotypes and diversified topotypic profiles of Foot-and-Mouth Disease Virus isolates in Ethiopia from 2007-2018”.

In this review article the authors try to summarize the FMD outbreak situation in Ethiopia during the period of 2007-2018. The topic is of great interest since understanding the situation of any particular region affected by FMD leads the pathway to control and eventually eradicate the disease. However, the manuscript is very poorly written and redundant over all, therefore is very difficult to follow and even to interpret what is the main massage of it all. Besides, there is barely any scientific data summarized in the text and some contradictory statements. For instance, the authors claim that Ethiopia is transected by 4 pools (line 278), and in the figure related to the text, figure 4, they show that the region is in the middle of 3 pools only.

Minor comets:

1.     Please, chose a way to present citations and be consistent throughout the entire manuscript.

2.     Likewise, in the Reference list, follow the journal format.

3.     The English should be reviewed thoroughly.